# Spatially Resolved Photo-Response of a Carbon Nanotube/Si Photodetector

**DOI:** 10.3390/nano13040650

**Published:** 2023-02-07

**Authors:** Daniele Capista, Luca Lozzi, Aniello Pelella, Antonio Di Bartolomeo, Filippo Giubileo, Maurizio Passacantando

**Affiliations:** 1Department of Physical and Chemical Science, University of L’Aquila, Via Vetoio, Coppito, 67100 L’Aquila, Italy; 2Department of Physics, University of Salerno, Via Giovanni Paolo II 132, Fisciano, 84084 Salerno, Italy; 3CNR-SPIN Via Giovanni Paolo II, Fisciano, 84084 Salerno, Italy; 4CNR-SPIN Via Vetoio, Coppito, 67100 L’Aquila, Italy

**Keywords:** carbon nanotubes, heterostructure, silicon heterostructure, photodetector, photodiode, photoconductivity, quantum efficiency

## Abstract

Photodetectors based on vertical multi-walled carbon nanotube (MWCNT) film-Si heterojunctions are realized by growing MWCNTs on n-type Si substrates with a top surface covered by Si_3_N_4_ layers. Spatially resolved photocurrent measurements reveal that higher photo detection is achieved in regions with thinner MWCNT film, where nearly 100% external quantum efficiency is achieved. Hence, we propose a simple method based on the use of scotch tape with which to tune the thickness and density of as-grown MWCNT film and enhance device photo-response.

## 1. Introduction

Photodetectors are devices that are able to convert an incoming optical signal into an electrical signal. They are widely used in our society, from in the automotive industry to biomedical and military applications. Cutting-edge devices try to achieve better performance by combining classical semiconductors with low-dimensional materials [1,2,3].

Heterostructures formed by one-dimensional materials such as carbon nanotubes (CNT) with silicon have gained more and more attention in recent years [4,5]. It is known that the outstanding chemical, mechanical, and electrical properties of CNTs make them suitable for many hybrid technological devices [6,7,8,9,10,11,12,13]. In particular, they are often used in combination with traditional semiconductor substrates to realize improved optoelectronic devices [14,15,16,17,18,19]. One way to realize such devices is by directly growing CNT above a silicon substrate through chemical vapor deposition (CVD) [20,21]. The electrical contact of CNT films with silicon generates a rectifying junction and, due to their high electrical conductivity and optical transparency, the nanotubes work both as an antireflective layer and conductive electrode for photo-charge collection [22,23,24].

In this study, we analyze the external quantum efficiency of silicon substrates covered by multi-walled carbon nanotube (MWCNT) film grown by CVD. Firstly, the electrical behavior of the device is tested in the dark and under a large-area light spot to characterize the junction behavior. Since the light spot entirely covers the nanotube surface, the obtained photocurrents are related to the position-averaged response of the detector. We then check how the photo-response of the device changed as a function of the light spot position. Afterward, we look at the sample morphology to correlate the electrical photo-response of the device with the nanotube distribution inside the MWCNT film. We observe that a lower thickness of MWCNT film corresponds to a higher photo-response. Hence, we develop a simple process to mechanically control the thickness of the CVD-grown CNT film. Using a piece of tape (normally used for the exfoliation of two-dimensional materials), a fraction of the nanotubes is removed from the surface of the device. The efficiency maps acquired after the MWCNT film-thinning process confirm the initial observation. The proposed thinning process therefore offers an innovative and easy way to improve the efficiency of MWCNT-Si photodetectors and thus represents a further step toward the exploitation of carbon nanotubes in real-life applications.

The main advantages of an MWCNT layer over a transparent metal contact are the higher transparency, electrical conductivity, and chemical stability.

## 2. Materials and Methods

The devices were realized starting from substrates purchased from MicroSens (Neuchâtel, Switzerland). The substrates were composed of a 500 µm n-type Si layer (resistivity 1–5 Ωcm, doping ~10^15^ cm^−3^) with the top surface covered by a nominal 140 nm thick Si_3_N_4_ layer (deposited via plasma-enhanced chemical vapor deposition) and the bottom surface covered by a 60 nm Pt-Ta layer. The substrate also presented two Pt-Ta pads of 1 mm^2^ each on the top side to facilitate the electrical measurements. The region of the MWCNT growth was selected by the deposition of a 3 nm thick Ni film on a 25 mm^2^ square region on the top surface of the substrate using thermal evaporation. After the Ni deposition, the substrates were taken into a chemical vapor deposition chamber that was pumped down to a pressure lower than 10^−5^ Torr. Successively, the substrates were annealed at 750 °C to enable the formation of MWCNTs. An NH_3_ gas flow at a rate of 100 sccm for 20 min was used to prevent Ni nanocluster oxidation. The annealing process transformed the Ni film into nanoclusters [25], which was necessary for the catalysis of the MWCNT growth. To allow the nanotube growth, C_2_H_2_ was added to the ammonia flow in the reaction chamber with a flow rate of 20 sccm for 10 min, retaining the same temperature as that of the annealing process. After MWCNT growth, the samples were slowly cooled down to room temperature.

For the electrical characterization, the devices were placed into a sample holder that allowed precise movements in the horizontal plane. The light coming from different LEDs (Ocean Optics, Ostfildern, Germany; with wavelengths: 380, 395, 470, 518, 590, and 640 nm) was directed above the surface of each sample using an optical fiber. It was possible to control the diameter of the light spot above the device by simply changing the distance between the optical fiber and the device surface. Independently of the wavelength of the LED radiation, the resulting light spot could be made as large as the MWCNT film to obtain the average photocurrent of the detector or made to have a minimum diameter of 1 mm to collect the photo-efficiency map.

Using a Keithley 236 source measure unit, we studied the current flowing between one of the top pads and the back of the devices. Firstly, a repeated series of electrical stresses was applied to a device until it showed reproducible behavior. The electrical stresses thinned the nitride layer below the nanotubes, allowing the formation of a metal–insulator–semiconductor (MIS) junction with the silicon nitride and the n-doped silicon. It is also assumed that an ohmic junction was formed on the back [26,27].

After electrical stabilization, the I–V characteristics of the device were acquired both in the dark and under a large light spot illumination. This initial characterization was followed by a series of measurements to obtain photo detection efficiency maps. All of the device’s surface was scanned by the small light spot, and the current was measured as a function of the spot position while a voltage (V_ph_) was applied between the top pad and the back of the device (Figure 1e). The photocurrent I_ph_ as a function of the light spot position was then calculated as I_ph_ = I_light_ − I_dark_ (where I_light_ is the current under illumination and I_dark_ is the current in the dark) and converted into quantum efficiency (QE) maps (through the relation QE = (I_ph_hc)/ePλ where P is the LED power and λ is the light wavelength).

We developed a procedure to reduce the density and thickness of the MWCNT film on selected areas. This procedure consisted of putting a piece of tape (normally used for 2D material exfoliation) above a portion of the device and then gently pressing it using a cotton swab. Part of the nanotube remained attached to the tape, and the removal of it left part of the device with a thinner MWCNT film (Figure 1a–c). Using a profilometer (Veeco, Tucson, AZ., USA; Dektak 6M), we measured the thicknesses of the two parts of the MWCNT film. The profiles taken along two lines, shown in Figure 1c, are reported in Figure 1d.

The morphologies of the MWCNT films of the devices before and after the removal process were analyzed using a field emission scanning electron microscope (SEM, Zeiss Leo 1530) at an accelerating voltage of 5 kV.

## 3. Results and Discussion

Figure 2a shows the I-V characteristic taken from one of the devices in the dark and under the light of a 380 nm LED. The dark curve shows that the device had a rectifying behavior, with an on/off ratio of 20 at ± 4V. The blue curve shows that the reverse current below −6 V started to increase exponentially when the device was under illumination. The linear behavior of the photocurrent in the Fowler–Nordheim plot in Figure 2b demonstrates that this increase was due to the tunneling of the photocharges through the triangular barrier of the MIS structure constituted by the MWCNT film, the silicon nitride, and the n-type silicon, as reported in our previous works [14,27]. The increasing reverse bias improves the separation and collection of photocharges, resulting in an increase in the current. This increase stops when an equilibrium between the generation and collection of photocharge is reached. After this point, the current does not grow further, and a plateau is reached.

We chose a value of V_Ph_ in the plateau region in reverse bias (Figure 2a) to obtain the quantum efficiency map. Figure 3a shows the efficiency map obtained from a device with a pristine MWCNT film. It is possible to see that the photo-response is different from zero only when the light spot is focused on the MWCNT film area. The efficiency of the device was almost constant along the MWCNT film, but along the top and bottom edges, it showed a higher response. To understand this behavior, we looked at the CNT film with a scanning electron microscope. Figure 3b shows the top-right corner of the MWCNT film. It can be observed that the film thickness on the upper edge slowly decreased until no nanotubes were present, while on the right edge, the thickness rapidly changed to zero. Comparing the map with the SEM images, it is clear that the regions with higher QE are associated with the regions with thinner MWCNT film.

To double-check the anti-correlation between film thickness and photo-response, we realized a second device wherein the film was mechanically thinned after CVD growth [28]. Figure 4a shows a SEM image of the MWCNT film after the thinning process. The nanotubes on the right are longer than the ones on the left. Figure 4b shows an optical image of the device after the thinning process. As reported in Figure 1d, the thickness of the MWCNT film was thinned by 80% of its initial value. Figure 4c,d compare the QE maps obtained before and after the thinning process. The QE was initially constant along all the MWCNT film area; however, after the thinning process, the efficiency in the thinner portion of the film showed a significant increase.

Using different LEDs, we evaluated the quantum efficiency of the two regions for different wavelengths. Figure 5a shows that the increment of efficiency was similar for the different wavelengths.

Figure 5b shows a comparison of the I-V curves taken using the left (L) or right (R) metallic pad as the electric contact in the three different conditions: dark and light focused on the pristine or thinned film. Looking at the dark currents of the device (black and grey dotted lines), it can be observed that for forward bias, the current was almost an order of magnitude greater when the contact was made on the right pad (the one on the pristine film). This could have been due to a lower contact resistance between the pristine MWCNT film and the metallic pads that directly affected the series resistance of the junction. Once illuminated, the response of the device changed accordingly with the contact used and the position of the spot above the film. Using the same power for all the measurements, the currents measured from the left pad (indicated by “L”) were always smaller than those measured from the right pads (indicated by “R”). Independently from the pad used for the measurements, it can be observed that the current plateau in reverse bias was reached at a lower bias when the spot was located above the pristine film, but its height was always lower than that reached when the spot was above the thinned film. It is therefore possible to recognize two regimes: one at a lower reverse bias where the photo-response is higher in the pristine film and one at a higher reverse bias where the thinned film is more efficient. To explain this phenomenon, it is necessary to consider how the photocurrent is generated: Firstly, the light must penetrate inside the device to generate electron-hole (e-h) pairs in the Si substrate; then, the e-h pairs need to be separated and collected through the electrical contacts to measure a photocurrent. Although the reduced thickness of the film enhances the photon flux to the Si substrate (and consequently the photocharge generation), on the other hand, the lower density of the nanotubes makes the process of photocharge collection less efficient.

The nanotubes constitute a semi-transparent metallic layer that work both as an antireflective layer and as a charge-collection film [29]. The thickness and density of the nanotubes need to be controlled to improve device performance.

The thinning process that we propose offers a very simple and cheap way to reduce the thickness and density of an MWCNT film directly grown on a substrate. The adhesive tape removes the highest nanotubes that shade the substrate, leaving only the shorter nanotubes. Thus, a higher number of photons can reach the silicon substrate and increase the quantum efficiency of the device. Other factors must be considered; the thinning process also increases the resistance of the film, increasing the voltage required to reach a plateau. Repeated thinning processes can also remove all the nanotubes from the substrate, nullifying the response of the device.

## 4. Conclusions

Photodetectors were realized by growing MWCNTs on the surface of an n-doped silicon substrate with surfaces covered by Si_3_N_4_ layers. The photo-responses of the devices were studied across their active surfaces to find possible differences in the local QE.

Comparing the results from the electrical measurements to the morphology of the MWCNT film, it was observed that a higher photo-response was associated with a lower nanotube film thickness. This observation was confirmed by devices where a portion of the film was thinned through a scotch tape exfoliation-like process. An efficiency map evidenced that a thinner MWCNT film resulted in a higher photocurrent.

The findings of this work indicate that MWCNT film acts both as a barrier that prevents light from reaching the substrate and as a conductive electrode for photo-charge collection. To attain the best device for photocurrent generation, the MWCNT film must be as large as possible to increase the active surface area and with a density of nanotubes that guarantees the electrical interconnection of the film and to minimize the dispersion of incoming light.

## Figures and Tables

**Figure 1 nanomaterials-13-00650-f001:**
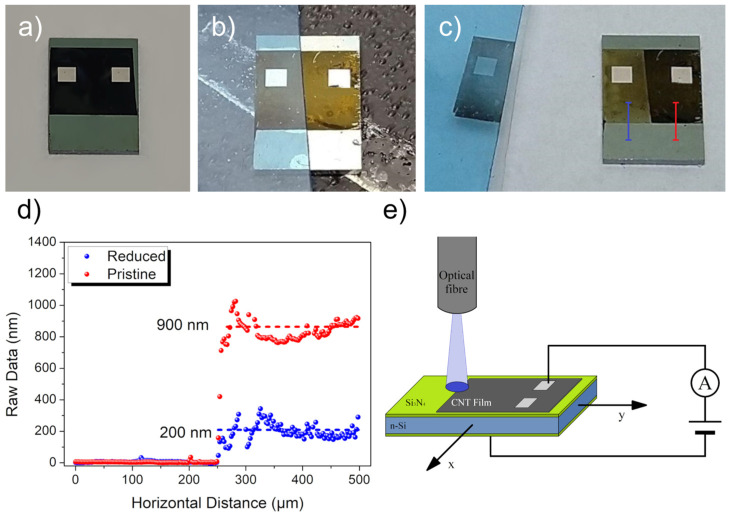
MWCNT film-thinning process: (**a**) device with pristine MWCNT film, (**b**) scotch tape adhered to half area of the device, and (**c**) tape peeled off from the device. (**d**) The thickness of the MWCNT film along the lines in (**c**). (**e**) The layout of the device and the setup used for the photo-response characterization as a function of the light spot position.

**Figure 2 nanomaterials-13-00650-f002:**
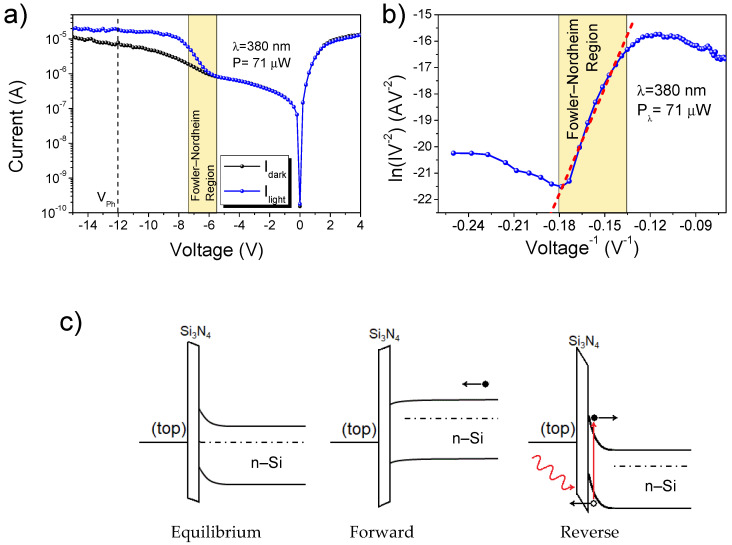
(**a**) I–V characteristic of the device in the dark and under illumination by a 380 nm LED. V_ph_ is the voltage used to estimate the photocurrent. (**b**) Fowler–Nordheim plot of the blue curve in (**a**). (**c**) Band structure of the MIS junction formed between the nanotubes, the silicon nitride, and the n-doped silicon at equilibrium, forward, and reverse bias. In the reverse bias condition, the photogenerated holes inside the silicon can tunnel through the triangular-shaped barrier of the silicon nitride, giving rise to a photocurrent.

**Figure 3 nanomaterials-13-00650-f003:**
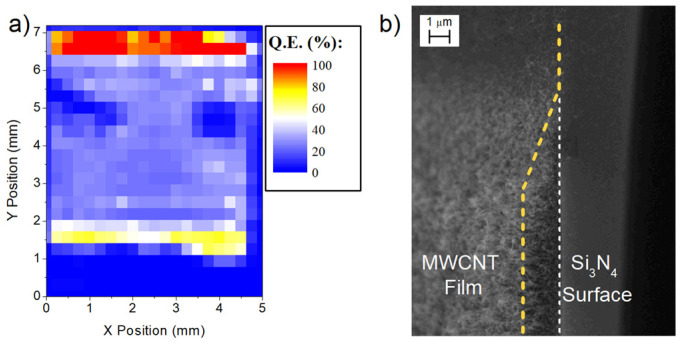
(**a**) Quantum efficiency map of a device (with pristine MWCNT film) that shows efficiency inhomogeneities across its surfaces. The device presents an average QE of 30%, but near the top and bottom edges, the QE increases to nearly 100%. (**b**) SEM images of the top-right corner of the MWCNT film. The film presents two types of edges: the top edge, which decreases gradually, and the right edge, which is sharper and akin to a step.

**Figure 4 nanomaterials-13-00650-f004:**
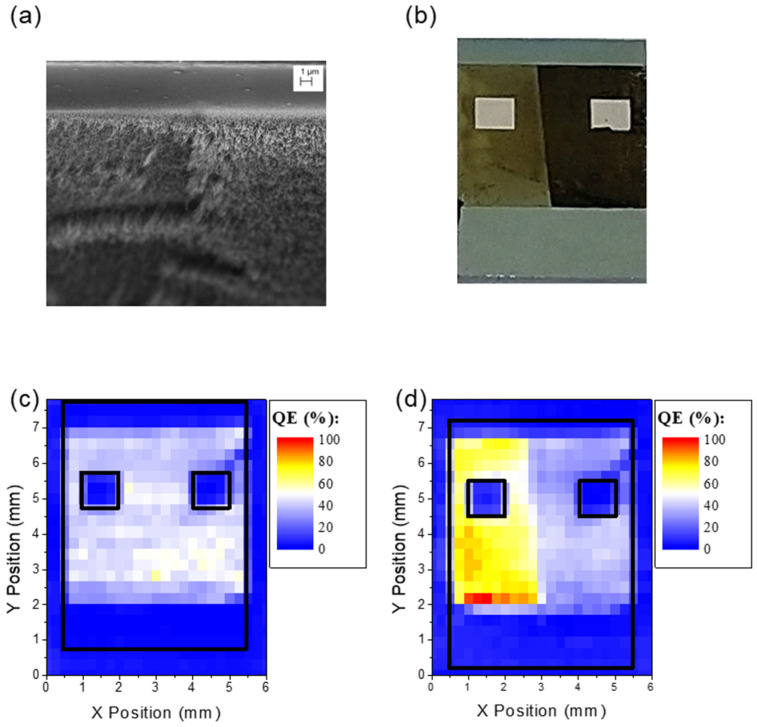
(**a**) SEM image of the device tilted by a 15-degree angle from the surface plane after the thinning process. (**b**) Optical image of the device after the thinning process. Quantum efficiency map of the device before (**c**) and after (**d**) MWCNT removal. The black lines mark the real dimensions of the substrate and metallic pads. Both maps were acquired using the right pad as an electrical contact.

**Figure 5 nanomaterials-13-00650-f005:**
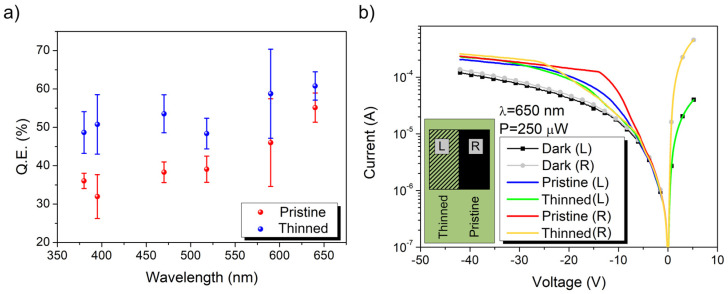
(**a**) Average QE of the device at the plateau and associated standard deviation (error bars) as a function of the wavelength. Red dots represent the values of the QE obtained on the pristine portion of the film, while the blue dots represent the values from the thinned portion. (**b**) Comparison of the I-V curves acquired using the left or right pads as electric contacts (indicated by the letters L and R), in the dark and under the light of a 650 nm laser at 250 µW (with light spot focused on the thinned or pristine film).

## Data Availability

Data available on request.

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
