# Peer review of "Spatially Resolved Photo-Response of a Carbon Nanotube/Si Photodetector"

_nanomaterials, 2023, doi:10.3390/nano13040650_

Round 1
Reviewer 1 Report
The authors use the idea to combine MWCNTs with Si for achieving high photoresponse mainly given by the photogeneration of carriers in the Si substrate.
Please consider the following comments:
- Line “Anti-correlation between film thickness and photo-response” – this was reported also by Appl. Phys. Lett. 119, 221108 (2021), https://doi.org/10.1063/5.0073355 for an ultrathin active layer of Ge QDs in SiO2 by increased IQE and responsivity compared to thin film
- Line 174 “The nanotubes work as a semi-transparent metallic layer” – surface photovoltage SPV contribution to photocurrent is facilitated as the film plays the role of conducting transparent electrode, please see Sci. Rep. 8, 4898 (2018), https://www.nature.com/articles/s41598-018-23316-3. In this way Si substrate has a beneficial role by increasing the photoconduction in the film by electrostatic doping - (related to “they are often used in combination with traditional semiconductor substrates to realize improved optoelectronic devices”).
Reviewer 2 Report
The paper is devoted to the study of the application of the MWCNT contact layer on the S3N4/Si structure to create high-performance photodiodes operating in the visible wavelength range. The effect of a significant increase in the external quantum efficiency (QE) of the photodiode (up to 80-100%) is found when the MWCNT layer is thinned to several hundreds of nanometers.
Unfortunately, despite the interesting results, there are several remarks to the article.
- Neither the title nor the abstract contains the abbreviation MWCNT, which makes it difficult to read and understand the material.
- Keyword recommendation. I would add something about Si, and also indicate photodiode or photodetector
- Line 36. The meaning of the term "average electrical behavior" is not entirely clear.
- The novelty of the study does not follow very clearly from the introduction. As I understand it, the authors claim that it consists in the use of MWCNT when creating heterostructures for a photodiode, in contrast to the already studied single-walled CNTs? Or is the novelty related to the method of preparation (CVD) and the scope of such MWCNT/Si structures? According to the work (Journal of Alloys and Compounds Volume 622, 2015, Pages 966-972 "Facile synthesis of uniform MWCNT@Si nanocomposites as high-performance anode materials for lithium-ion batteries" https://doi.org/10.1016/j .jallcom.2014.11.032) such structures have already been grown, but by a different method and for different purposes. It is extremely important to clearly indicate the novelty of your research, and in all categories (object, method of obtaining, scope, observed phenomena, etc.).
- Line 51. What method was used to grow the S3N4 layer? How was the layer thickness controlled (in situ and/or ex situ)?
- Line 51. The figure of speech "The substrate also presented Pl-Ta film" is not entirely clear. Do you mean that the substrate was covered with a layer of Pl-Ta?
- Line 52. It is not clear what size and shape the Ni-covered MWCNT growing region was.
- Line 53-54. The turn of phrase "After the evaporation, the substrates were taken into a Chemical Vapor Deposition chamber" is not very good. This can be read as substrate evaporation, while the authors obviously mean the prior procedure of thermal evaporation.
- Line 57-58. Is the formation of Ni nanoclusters confirmed by the experimental data of the authors? If not, then it is necessary to provide confirmation in the form of a reference to other works.
- Line 58. Do I understand correctly that during the growth of CNT there was both NH3 and C2H2 in the chamber? If so, could this somehow affect the composition of the CNT?
- How did the authors control the formation process to get exactly MWCNT, as opposed to single walled CNT? This is not explicitly stated in the section on the growth of structures.
- Line 62-63. Probably, the sentence is not very correct in terms of word order in English.
- Line 64. The wavelengths of the LEDs used are not indicated. Is there a dependence of the size of the light spot on the wavelength of the LED radiation?
- It is not very clear how the photocurrent flows in the described structure. The authors write that measures were taken to thin Si3N4, which led to the formation of contacts. Is it known what thickness of Si3N4 eventually formed under the CNT? It would be very desirable to present and discuss the band diagram of the instrument and to discuss the mechanism of photocurrent flow.
- Line 70. The abbreviation MIS is not clarified.
- It is not explicitly stated which structure's QE measurement results are shown in Figure 3. (Apparently this is not a thinned structure).
- Authors use the designations MWCNT and CNT equally often to refer to the same thing. It is desirable to bring to uniformity. As I understand it, in this work all structures with MWCNT.
- Line 134. Perhaps the authors made a typo and need a link to Figure 1d?
- The authors do not explicitly indicate what factors make up the error in determining QE in Figure 5a.
- It is not clear the difference between the L and R parts of the structure and pristine and reduced, which are shown in Figure 5b. As I understand it, the L and R parts differ in the conduct / non-thinning procedure. So, L and R are the same as reduced and pristine? What then do the blue and yellow lines in the figure mean?
- An explanatory picture is required, which will show the mechanism for generating photocurrent.
- As I understand from the text, the conversion of light into e-h pairs occurs in Si, and the MWCNT layer is a contact layer. It is not clear why this is better than a conventional metal transparent contact.
Round 2
Reviewer 2 Report
The authors responded satisfactorily to all comments except â„–9, where they are not consider probalbly affect of NH3 in MWCNT composition. It should be discussed.
